# H$_2$ formation via non-Born-Oppenheimer hydrogen migration in photoionized ethane

Yizhang Yang[1,8], Hao Ren[2,8], Ming Zhang ⬚[3,8], Shengpeng Zhou[1], Xiangxu Mu[3], Xiaokai Li[1], Zhenzhen Wang ⬚[1], Ke Deng[1], Mingxuan Li[1], Pan Ma[1], Zheng Li ⬚[3,4,5] ✉, Xiaolei Hao ⬚[2] ✉, Weidong Li[6], Jing Chen ⬚[6,7], Chuncheng Wang ⬚[1] ✉ & Dajun Ding ⬚[1] ✉

Neutral H$_2$ formation via intramolecular hydrogen migration in hydrocarbon molecules plays a vital role in many chemical and biological processes. Here, employing cold target recoil ion momentum spectroscopy (COLTRIMS) and pump-probe technique, we find that the non-adiabatic coupling between the ground and excited ionic states of ethane through conical intersection leads to a significantly high yield of neutral H$_2$ fragment. Based on the analysis of fingerprints that are sensitive to orbital symmetry and electronic state energies in the photoelectron momentum distributions, we tag the initial electronic population of both the ground and excited ionic states and determine the branching ratios of H$_2$ formation channel from those two states. Incorporating theoretical simulation, we established the timescale of the H$_2$ formation to be ~1300 fs. We provide a comprehensive characterization of H$_2$ formation in ionic states of ethane mediated by conical intersection and reveals the significance of non-adiabatic coupling dynamics in the intramolecular hydrogen migration.

Ultrafast migration and elimination of hydrogen in molecules is a prototypical reaction, which can be initiated in many ways, including electron impact ionization, X-ray absorption and multiphoton ionization[1–6]. Being the most abundant interstellar molecule, neutral H$_2$ serves not only as an important reactant for the synthesis of H$_3^+$ and complex hydrocarbon molecules, but also as an essential product resulting from the ionization reactions of hydrocarbon molecules, thus it plays an important role in the circulation of various molecular constituents within interstellar clouds[7–11]. Uncovering the mechanism behind neutral H$_2$ formation will advance our understanding of the complex reactions related to hydrogen migration in astrochemistry and ultrafast molecular processes. Especially, in the large amplitude dynamics of light atoms in molecules, such as hydrogen atom, both nuclear and electronic quantum effects can be prominent. Extensive researches have been conducted to elucidate the underlying mechanisms of the formation of ionic products, such as H$_2^+$ and H$_3^+$, where the dynamics of neutral H$_2$ play a key role[12–17]. Probing the dynamics of neutral H$_2$ formation from the ionized molecules poses challenges due to its ultrafast and large amplitude motion in femtosecond timescale as well as its relatively high

[1]Institute of Atomic and Molecular Physics and Jilin Provincial Key Laboratory of Applied Atomic and Molecular Spectroscopy, Jilin University, 130012 Changchun, China. [2]Institute of Theoretical Physics and Department of Physics, State Key Laboratory of Quantum Optics and Quantum Optics Devices, Collaborative Innovation Center of Extreme Optics, Shanxi University, 030006 Taiyuan, China. [3]State Key Laboratory for Mesoscopic Physics and Frontiers Science Center for Nano-Optoelectronics, School of Physics, Peking University, 100871 Beijing, China. [4]Collaborative Innovation Center of Extreme Optics, Shanxi University, 030006 Taiyuan, Shanxi, China. [5]Peking University Yangtze Delta Institute of Optoelectronics, 226010 Nantong, Jiangsu, China. [6]Shenzhen Key Laboratory of Ultraintense Laser and Advanced Material Technology, and College of Engineering Physics, Shenzhen Technology University, 518118 Shenzhen, China. [7]Hefei National Research Center for Physical Sciences at the Microscale and School of Physical Sciences, Department of Modern Physics, University of Science and Technology of China, 230026 Hefei, China. [8]These authors contributed equally: Yizhang Yang, Hao Ren, Ming Zhang. ✉e-mail: zheng.li@pku.edu.cn; xlhao@sxu.edu.cn; ccwang@jlu.edu.cn; dajund@jlu.edu.cn

ionization potential. Theoretical studies of the reaction pathways for $H_2$ formation from hydrocarbon molecules have predominantly been conducted within the adiabatic Born–Oppenheimer approximation[18,19]. However, considering the high mobility of light hydrogen atoms, the non-adiabatic coupling of different electronic states beyond the Born–Oppenheimer approximation is expected to be prominent, for example, Mi et al. revealed the breakdown of Born–Oppenheimer approximation in the aotoionisation of the simplest molecule $H_2$[20]. The fast and large amplitude motion of hydrogen can induce strong non-adiabatic coupling between the ground and excited ionic states through conical intersections (CIs) in complex hydrocarbon molecules, which can proceed within a few femtoseconds time scale[21–25]. Despite the significance of non-adiabatic coupling between electronic states in the excited state dynamics of polyatomic molecules[26–31], the mechanism underlying ultrafast $H_2$ formation via hydrogen migration beyond the Born–Oppenheimer approximation remains poorly understood.

Ethane is a prototypical molecule for investigating ionization-induced neutral $H_2$ formation. After being irradiated by photons or electrons, the dominant fragmentation pathway involving neutral $H_2$ formation is $C_2H_6^+ \rightarrow C_2H_4^+ + H_2$. Experimental observations indicate that the yield of neutral $H_2$ is significantly higher than what is predicted by ab initio calculations based on the Born–Oppenheimer approximation, suggesting the crucial role of non-adiabatic coupling between electronic states in this process[18,19,32]. To characterize the dynamics of $H_2$ formation, it is necessary to employ experimental techniques capable of disentangling the non-adiabatically coupled electronic states and resolving ultrafast electronic nuclear dynamics with femtosecond temporal resolution. Thus we employed a two-dimensional channel-resolved orbital symmetry- and energy-sensitive electron spectroscopy with the cold target recoil ion momentum spectroscopy (COLTRIMS)[33–35] (detailed in Supplementary Note 1).

In this work, we resolve the contributions of the ground and excited ionic states to the $H_2$ formation channel. Moreover, we determine the branching ratios of $H_2$ formation from the coupled states mediated by the CI, as well as the coherent nuclear modes that constitute the branching space of the CI and the timescale of the non-adiabatic $H_2$ formation dynamics.

## Results
### Two-dimensional channel-resolved orbital symmetry- and energy-sensitive electron spectroscopy

The ground state electronic configuration of ethane is $(1a_{1g})^2(1a_{1u})^2(2a_{1g})^2(2a_{1u})^2(1e_u)^4(3a_{1g})^2(1e_g)^4$. The highest occupied molecular orbital (HOMO) $1e_g$ orbitals possess C-H bonding and C-C anti-bonding characteristics (see Supplementary Figure 1), and ionizing an electron from HOMO orbital will populate the $^2E_g$ state. On the other hand, the $3a_{1g}$ (HOMO-1) orbital exhibits C-C and C-H bonding characteristics, and ionizing from this orbital will lead to population of the dissociative $^2A_{1g}$ state strongly due to lowered bonding order[36]. In our experiment, an intense femtosecond laser pulse with a central wavelength of 800 nm and an intensity of 80–90 TW/cm$^2$ is employed to trigger the ionization of neutral ethane molecule, which is followed by the formation of $H_2$. The energy gap between $^2A_{1g}$ and $^2E_g$ states is ~0.7 eV[37], which is much smaller than the photon energy. As a result, both strong field tunneling ionization (TI) from the HOMO and HOMO-1 orbitals can be significant[38], leading to population of the $^2E_g$ and $^2A_{1g}$ cationic states within a sub-cycle timescale[39–42]. In our experiment, the Keldysh parameter $\gamma \sim 1$, indicating that the cationic states can also be populated through Freeman resonance (FR), where the high-lying neutral Rydberg states are initially excited via resonant multiphoton absorption, and subsequently ionized by absorbing one another photon[43–45].

The initial populations of the $^2E_g$ and $^2A_{1g}$ states through TI or FR and the subsequent relaxation pathways of the ethane cation are depicted in Fig. 1a. The calculation shows that the two coupled states can both contribute to the $H_2$ formation channel, where the dashed arrows indicate the electronic state switching pathways mediated the CI (see Fig. 1b). Figure 1c depicts a representative geometry for the CI between the two crossing electronic states, which reveals the transition state geometry for $H_2$ formation. The vectors **g** and **h** represent the gradient difference vector and linear coupling vector, respectively, forming the branching space of the CI[26] (see Supplementary Note 2 for details of the CI geometry). To distinguish the individual roles of the coupled $^2E_g$ and $^2A_{1g}$ states, we used the COLTRIMS (see Methods and Supplementary Note 1) technique and measured the photoelectron momentum distributions (PMDs) in coincidence with the parent ion ($C_2H_6^+$) and $H_2$ formation ($C_2H_4^+ + H_2$) channels at a laser intensity of

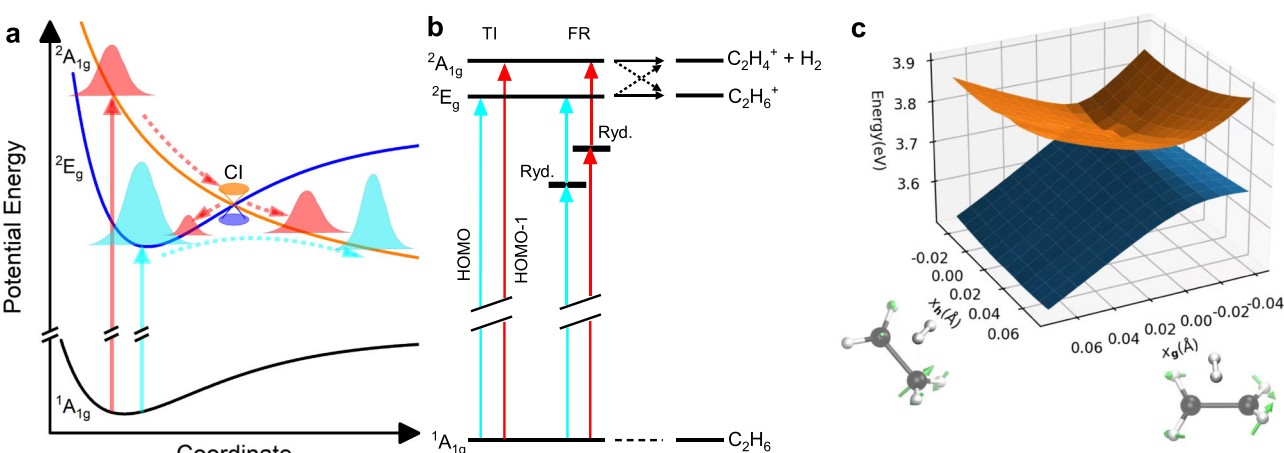

**Fig. 1 | Schematic for the $H_2$ formation reaction of $C_2H_6^+$. a** $H_2$ formation contributed both via direct dissociation from the excited $^2A_{1g}$ state and non-adiabatic coupling from the ground $^2E_g$ state via a conical intersection (CI). **b** The pathways of tunneling ionization (TI), Freeman resonance (FR) ionization processes and the following dissociation involving the non-adiabatic coupling between the $^2E_g$ and $^2A_{1g}$ states. These states can be populated through TI by directly removing an electron from the HOMO or HOMO-1 orbital, respectively. Alternatively, they can

be populated through FR, where additional photons are absorbed from different intermediate Rydberg states (Ryd.). **c** A representative CI geometry shows the character of $H_2$ formation and the potential energy surfaces along the gradient difference vector **g** and linear coupling vector **h**. The **g** and **h** vectors, represented by green arrows, define the branching space of the CI. Black and white spheres represent (C and H) atoms respectively.

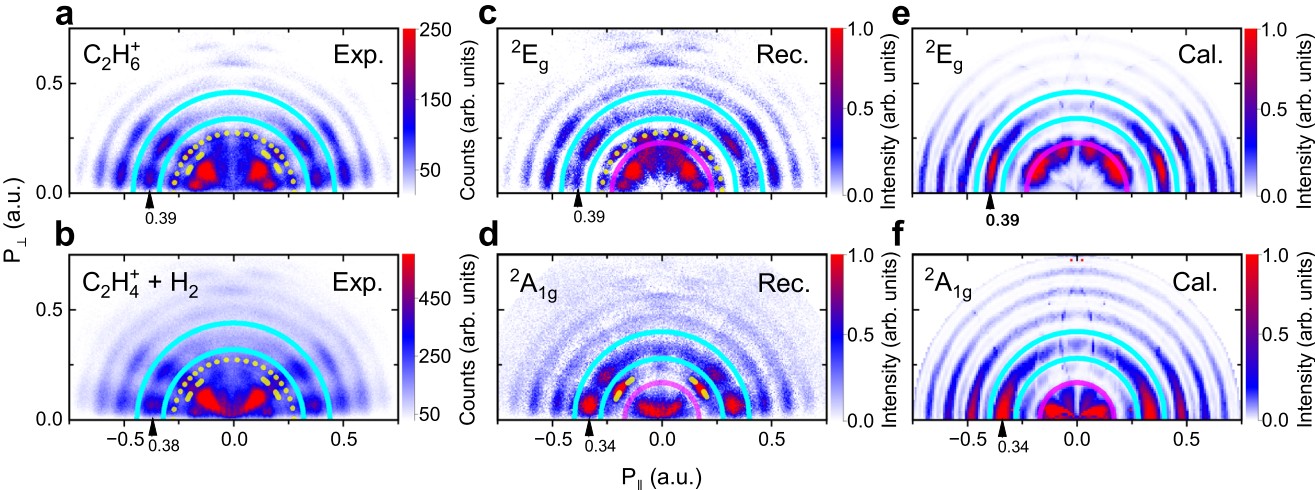

**Fig. 2 | Reconstructed and calculated PMD corresponding to the $^2E_g$ and $^2A_{1g}$ states of ethane. a, b** The measured photoelectron momentum distributions (PMDs) in coincidence with the parent ion ($C_2H_6^+$) and $H_2$ formation ($C_2H_4^+ + H_2$) channels in a laser intensity of 82 TW/cm². The yellow dashed lines indicate structures arising from two FRs. **c, d** The PMDs reconstructed from experimental data, which correspond to the $^2E_g$ and $^2A_{1g}$ states of ethane. **e, f** Theoretical calculation of the PMDs for the $^2E_g$ and $^2A_{1g}$ states in the same laser intensity, where the contributions from different molecular orientations are averaged. In (**c**)–(**f**), the electronic momentum distribution within the pink semicircle represents the fanning out stripes or arm-like structure for ethane, while the above-threshold ionization (ATI) within the blue solid lines represents the jet-like structures.

82 TW/cm² (Fig. 2a, b). The two-dimensional PMD provides unique information about the energy and symmetry of the ionized orbital, enabling to resolve dynamics in the different ionic electronic states. In Fig. 2, the PMDs exhibit signatures of above-threshold ionization (ATI) rings (within blue solid lines) for both channels in Fig. 2a, b, and the energy spectra of ATI show well-resolved comb structures (see Supplementary Fig. 3). The measured PMDs for the parent ion and $H_2$ formation channels contain two components contributed by the $^2E_g$ and $^2A_{1g}$ states.

In order to disentangle the contributions from the two coupled states to the PMDs of the $H_2$ formation and parent ion channels, we first fit the angular-integrated ATI spectrum of the parent ion and $H_2$ formation channels with two series of individual peaks spaced by one-photon energy. The relative energy offset between the two components matches the ionization energy gap between the two electronic states, which is ~0.7 eV, as sketched in Supplementary Fig. 3. These two components can be assigned to the ATI of the $^2E_g$ and $^2A_{1g}$ states, respectively. Thus we can determine the relative contribution of the $^2E_g$ or $^2A_{1g}$ state to the parent ion and $H_2$ formation channels (see details in Supplementary Note 3). Based on these results, we conclude that the measured PMDs of parent ion and $H_2$ formation channels (Fig. 2a, b) are formed by a linear combination of the contributions from the $^2E_g$ and $^2A_{1g}$ states with different relative ratios. Secondly, considering that the angular-dependent TI yields of the molecule lead to different relative ratios of the two components for various electron emission angles, we divided the PMDs into six regions. By fitting each integrated ATI spectrum in these regions using the same approach as in the previous step, we determine the relative ratios of the two components. The values of relative ratios from the $^2E_g$ state to the parent ion channel ($B_1$%) and $H_2$ formation channel ($B_2$%) for all regions are listed in Supplementary Table 2. With these ratios, the PMD solely from the $^2A_{1g}$ state can be reconstructed by subtracting the normalized PMD of the parent ion channel (multiplying $B_2$%) from that of $H_2$ formation channel (multiplying $B_1$%). Similarly, we can obtain the PMD from the $^2E_g$ state (see more details in Supplementary Note 4). The reconstructed PMD solely from the $^2E_g$ or $^2A_{1g}$ state is presented in Fig. 2c, d, respectively (see Supplementary Note 4 for detailed reconstruction procedures). To validate the reconstructed state-resolved PMDs of the $^2E_g$ and $^2A_{1g}$ electronic states, we employed the Coulomb-corrected strong-field approximation (CCSFA) method[46] to calculate the PMDs

from TI, which are correlated with the $^2E_g$ and $^2A_{1g}$ states (further details of CCSFA approach are presented in Supplementary Note 5). The calculated state-resolved PMDs are shown in Fig. 2e, f and compared with the PMDs reconstructed from experimental data, demonstrating excellent consistency.

In these PMDs, we observed distinct angular patterns, including jet-like structures (within the solid blue curves in Fig. 2) and fanning-out stripes (within the pink semicircle). These patterns can be attributed to either multiphoton ionization or sub-cycle electron wavepacket interference, which has been discussed previously[47,48]. Comparing the PMDs reconstructed from the measurements with those obtained from calculations, we found quantitative agreement in terms of the jet-like structure of the ATI ring and the low-energy structures. Further validation is conducted through a detailed comparison of their one-dimensional angular distribution (see Supplementary Note 4). In the case of HOMO ionization leading to the $^2E_g$ state, the fanning-out structure contains six stripes, which correspond to the 11-photon channel of multiphoton ionization, while the jet-like structure originating from the 12-photon channel of ATI exhibits seven nodes. And the calculations yield the same number of stripes and nodes. For HOMO-1 ionization resulting in the $^2A_{1g}$ state, the jet-like structure from the 12-photon channel of ATI also contains seven nodes, which is consistent with the calculation. Interestingly, the low-energy region in the reconstructed and calculated PMDs of the $^2A_{1g}$ state both exhibit an arm-like distribution, which significantly differs from that of the $^2E_g$ state. Applying the same procedure, we reconstructed the PMDs corresponding to the $^2E_g$ and $^2A_{1g}$ states from the experimental data at a higher intensity of 88 TW/cm². Their characteristic patterns in the very low energy region agree with Fig. 2c, d and are well reproduced in the calculations (see Supplementary Note 4). From the reconstruction of the electronic state-resolved PMDs, it becomes evident that the PMD of the $H_2$ formation channel can only be reproduced by considering the contributions from both the $^2E_g$ and $^2A_{1g}$ states. This finding strongly suggests that the non-dissociative $^2E_g$ state can contribute to the yield of the $H_2$ formation channel via non-adiabatic coupling-induced population transfer to the dissociative $^2A_{1g}$ state.

As shown in Fig. 2c–f, the most prominent orbital symmetry-dependent features lie in the low-energy region within the pink semicircle: the fanning out structure for the $^2E_g$ state and the arm-like structure for the $^2A_{1g}$ state. According to the CCSFA theory, the orbital-

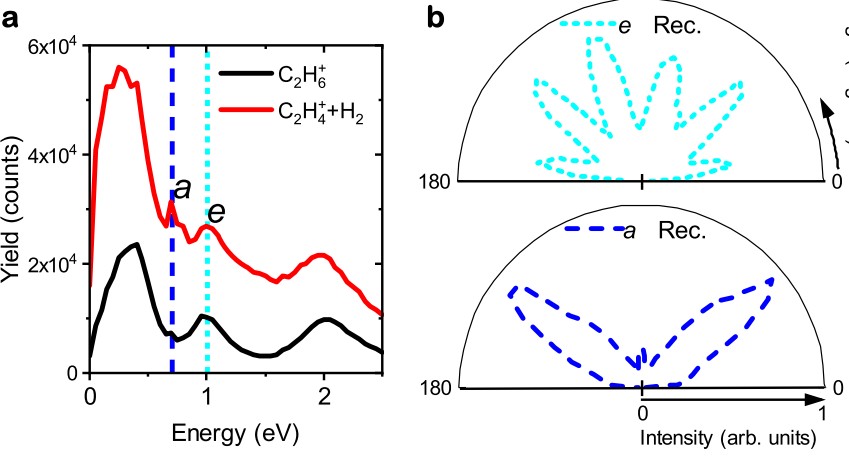

**Fig. 3 | Energy and angular distributions of electrons from Freeman resonance.**
**a** ATI spectrum of the parent ion and $H_2$ formation channels for electron ejection angles of 40(±10)° and 140(±10)° relative to the laser polarization. The spectrum reveals distinct features, including the sharp FR peaks at 1.0 eV (labeled as peak $e$)

and 0.7 eV (labeled as peak $a$), indicated by cyan and blue dashed lines, respectively. **b** The six-node structure of peak $e$ and the two-dominant-node structure of peak $a$ in the two-dimensional PMDs reconstructed in Fig. 2c, d.

dependent characteristics of the PMDs are rooted in the symmetry of the wave function of the molecular orbitals. Because the PMD is affected by the symmetry of molecular orbitals, ionizing from the antibonding character dominated $e_g$ orbital (HOMO) of odd mirror symmetry and the bonding character dominated $a_{1g}$ orbital (HOMO-1) of even mirror symmetry results in electronic state-sensitive PMD signatures[37,49] (further explanation can be found in Supplementary Note 5 and Supplementary Figure 7). Consequently, the distinct arm-like structure in the PMD serves as a symmetry-sensitive fingerprint of $a_{1g}$ orbital ionization, which was utilized to extract the population of the $^2A_{1g}$ state in our approach.

Apart from the TI, ionization via FR can also populate the $^2E_g$ and $^2A_{1g}$ cationic states and leave footprints in the PMD (see the indication of dashed lines in Fig. 2a, b). The characteristic angular distribution of photoelectrons resulting from FR ionization in the reconstructed PMD is located between the pink semicircle and solid blue lines in Fig. 2c, d, respectively. These structures are absent in the calculated PMDs since the FR was not taken into account in the CCSFA theory. Figure 3a shows the measured energy spectra for electron ejection angles of 40 (±10)° and 140 (±10)° relative to the laser polarization, with the dashed lines indicating the sharp peaks corresponding to FR at 1.0 eV (peak $e$) and 0.7 eV (peak $a$). Peak $a$ contains two dominant nodes along 40° and 140°, while peak $e$ displays six nodes (as seen in Fig. 3b), which are also evident in Fig. 2a–d. The characteristic angular distributions of peak $a$ and peak $e$ could be attributed to FR mediated by two neutral Rydberg states with different symmetries, i.e. with different orbital angular momentum quantum numbers. Moreover, peak $a$ is more pronounced in the dissociative $H_2$ formation channel relative to the parent ion channel, indicating that the initial ionization pathway producing peak $a$ is from the Rydberg state to the dissociative cationic state $^2A_{1g}$. In contrast, peak $e$ is more prominent in the parent ion channel, suggesting that the initial ionization pathway for peak $e$ is from the Rydberg state to the stable ground ionic state $^2E_g$. The ionization pathways for peak $a$ and peak $e$ are illustrated as vertical red and cyan arrows, respectively, in Fig. 1. Once the $^2E_g$ and $^2A_{1g}$ states are populated, non-adiabatic molecular dynamics (MD) is initiated, which is similar to the dynamics after TI.

## Branching ratios of $H_2$ formation channels for two coupled electronic states
By extracting the populations of $^2E_g$ and $^2A_{1g}$ states with the orbital symmetry-dependent features through TI and FR, we determined the branching ratios of parent ion and $H_2$ formation channels for two

coupled cationic states, which is crucial to comprehend the non-adiabatic relaxation dynamics through CI. With the single-event coincidence measurement, we collected the total counts of electrons corresponding to the parent ion channel ($N_P$) and the dissociative $H_2$ formation channel ($N_D$) as $N_P = 22300$ and $N_D = 58041$ for a laser intensity of 88 TW/cm$^2$. Considering the respective weights of the $^2E_g$ in the $H_2$ formation (48%) and parent ion channels (69%), the total counts of electrons populated in the $^2E_g$ state can be calculated as $N_E = 0.69N_P + 0.48N_D = 43247$. Similarly, the total counts for $^2A_{1g}$ state are calculated to be $N_A = 37094$. The branching ratio of the $H_2$ formation channel in the $^2E_g$ state can then be calculated as $0.48N_D/N_E = 0.64$. Similarly, the branching ratio of the parent ion channel for the $^2A_{1g}$ state can be derived and we displayed them in Fig. 4. We found that 19(±9)% of the $^2A_{1g}$ state population relaxes to the parent ion channel, while 64(±8)% of the $^2E_g$ state population is directed towards the $H_2$ formation channel. We illustrate the branching ratios for two peak laser intensities in Fig. 4, and listed them in Supplementary Table 5.

On the other hand, we derived the branching ratios of cations produced by the FR ionization from the collected counts, which is independent of the fitting procedure used for TI. For the FR ionization, the branching ratio of $H_2$ formation channel in the $^2E_g$ state equals to the counts of peak $e$ in coincidence with dissociative $H_2$ formation with $C_2H_4^{2+}$+$H_2$ fragments ($N_{eD} = 14337$) divided by the total counts of peak $e$ for both $H_2$ formation and parent ion channels ($N_e = 19901$), which is 72(±4)% in the case of 82 TW/cm$^2$. Similarly, the parent ion channel from the $^2A_{1g}$ state has $N_{aP} = 764$, $N_a = 4005$ and the branching ratio equals to $N_{aP}/N_a = 19(±8)$%. All the branching ratios were compared in Fig. 4 and their values were found to be consistent with those obtained from TI for both laser intensities (see details in Supplementary Note 6). We calculated the branching ratios of two channels using the non-adiabatic surface hopping MD simulation on the CASSCF(5,5)//6-31G* level (see Supplementary Note 2), and the results shown in Fig. 4 are in consistency with the experimental measurement, which provides solid validation to our interpretations. Thus we demonstrated that more than 60% of initial populations of the $^2E_g$ state and more than 70% of populations of the $^2A_{1g}$ are relaxed to the $H_2$ formation channel, which explains the high $H_2$ yield from photoionized ethane in the experiment.

## Time-resolved neutral $H_2$ formation dynamics beyond Born–Oppenheimer approximation
To characterize the dynamical effect of non-adiabatic coupling in $H_2$ formation process, we performed femtosecond time-resolved ion

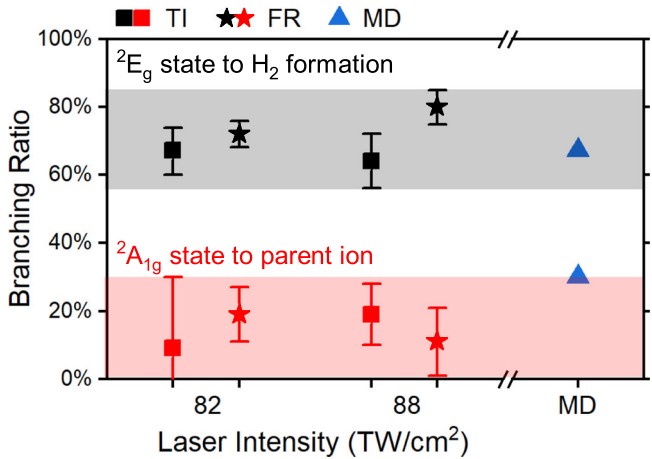

**Fig. 4 | Branching ratios of H₂ formation channel after the CI.** The branching ratios of $H_2$ formation channel in the $^2E_g$ state and the parent ion channel in the $^2A_{1g}$ state are represented by black and red colors, respectively. The contributions from TI and FR processes are denoted by squares and stars. The branching ratios of TI and FR processes at two different laser intensities (82 and 88 TW/cm²) are compared, and their average values are indicated by a dashed line. The absolute laser intensity calibration error is approximately 15%, while the relative intensity between two conditions is 6(±1) TW/cm² (see Methods). The shaded areas stand for the maximum uncertainty ranges of the branching ratios. The calculated branching ratios for each channel using non-adiabatic molecular dynamics (MD) simulation are presented as blue triangles. The errors displayed for TI and FR are propagated from the fitting errors and statistical uncertainties, respectively.

yield measurements in order to track the ultrafast hydrogen motion on real time and obtain the intrinsic timescale for $H_2$ formation. A pump pulse is used to prepare $C_2H_6^+$ and initiate the nuclear wavepacket motion in cationic states, and since the production of $C_2H_4^+$ and neutral $H_2$ is a concentrated process, the $H_2$ formation dynamics can be monitored by ionizing $C_2H_4^+$ to $C_2H_4^{2+}$ at different time delays with a probe pulse. Figure 5a presents the $C_2H_4^{2+}$ yield as a function of pump-probe time delay. The yield initially reaches a maximum at time zero as the pump and probe pulses overlap and then rapidly decreases. However, after a time delay of approximately 700 fs, the yield starts to rise again, as shown in the inset. This exponential rise is accompanied by oscillations in the yield, indicating the involvement of non-adiabatic coupling between the $^2A_{1g}$ and $^2E_g$ states in the $H_2$ formation dynamics. As magnified in Fig. 5a, the yield rising with pump-probe time delay ($y(t)$) can be well fitted by an exponential function $y(t) = A[1 - e^{-(t-t_0)/\tau}]$, where $A$ is the amplitude, $t_0 = 748(\pm 24)$ fs is the initial instant of the rise and $\tau = 567(\pm 61)$ fs is the time duration for the yield rising process. Surprisingly, the scale of this time duration (approximately 1300 fs) is much longer than that of the $H_2^+$ or $H_3^+$ formation (approximately 200 fs) observed in the cationic or dicationic states of other hydrocarbon molecules[10,27], which indicates their dramatically different temporal features for the underlying non-adiabatic dynamics of $H_2$ formation.

Moreover, the oscillations accompanying with yield rising provide the fingerprints of the coherent vibrational wavepacket motion during the $H_2$ formation. The results of Fourier transform analysis for the residual signals are shown in Fig. 5b. The Fourier transform of experimental data shows four peaks between 100 cm⁻¹ and 200 cm⁻¹, which are assigned to the vibrational frequencies of the representative CI geometry shown in Fig. 1c (see Supplementary Note 2 for the vibrational modes of the CI geometry). Other frequencies can be assigned to the frequencies of $C_2H_6^+ \rightarrow C_2H_4^+ + H_2$ transition state geometry[18]. To demonstrate the relationship between the emergence of these vibrational frequencies and the dynamics of $C_2H_6^+$ evolving to the CI geometries with $H_2$ formation character, we define an overlap of the continuous wavelet transform (CWT) time-resolved frequency

distribution of $C_2H_4^{2+} + H_2$ yields oscillation $I_{CWT}(\omega, t)$ and the characteristic frequency of CI $I_{CI}(\omega)$ (defined in SM Section II), $I_{overlap}(t) = \int_0^{\omega_{max}} I_{CI}(\omega) I_{CWT}(\omega, t) d\omega$, where $\omega_{max} = 300$ cm⁻¹ is the cutoff frequency. As shown in Fig. 5c, d, the overlap $I_{overlap}(t)$ from experimental frequency analysis data reaches a maximum at 1234 fs. This implies that $C_2H_6^+$ evolving to the CI geometry takes a time of more than 1200 fs, which is consistent with the result from velocity autocorrelation function of MD trajectories reaching a maximum at 1286 fs.

## Discussions

The $H_2$ formation dynamics can also be monitored by ionizing $H_2$ to $H_2^+$ ($C_2H_4^+ + H_2^+$ channel). Based on the ion-ion coincidence technique, the delay time-dependent kinetic energy release (KER) of the ($C_2H_4^+ + H_2^+$) channel is presented in Supplementary Figure 10. The appearance of a band with low KER after approximately 700 fs indicates that the nuclear wavepacket starts to reach the CI. As the time delay increases from 700 fs, the yield shows a clear rise, consistent with the ($C_2H_4^{2+} + H_2$) channel as expected. At the same time, its KER decreases from approximately 1 eV to 0.5 eV, indicating that the distance between $C_2H_4^+$ and $H_2$ increases as the delay time increases.

Moreover, at earlier time delays before the formation of neutral $H_2$, the initial nuclear wavepacket prepared by the pump pulse can be projected onto higher excited states by absorbing more photons from the probe pulse. This process can lead to the formation of a dissociative channel: $C_2H_6^+ \rightarrow CH_3^+ + CH_3$. The ionization yield of this channel (Supplementary Fig. 11) exhibits an enhancement after approximately 500 fs and starts to decrease after approximately 1300 fs. The time-dependent yield of this channel is opposite to that of the $C_2H_4^{2+} + H_2$ channel, suggesting that the two channels compete with each other. For early times between 500 fs and 1300 fs, the dissociative channel induced by photoexcitation plays an important role; for longer delay times, the $H_2$ formation channel becomes dominant.

In conclusion, we revealed that the neutral $H_2$ formation dynamics via intramolecular hydrogen migration goes beyond the Born−Oppenheimer approximation for the cationic ethane. Based on the energy- and orbital symmetry-sensitive photoelectron coincidence spectroscopy, the two coupled electronic states can be quantitatively disentangled, and we obtained the state-resolved branching ratios for $H_2$ formation. We further found that the nuclear wavepacket takes ∼1300 fs to reach to the CI, which leads to the enhanced yield of the $H_2$ formation channel. Analysis of the observed vibrational fingerprints provided the vibrational modes forming the branching space of the CI. Our work demonstrates the critical role of non-adiabatic effect in the $H_2$ formation, which can help to interpret the large amplitude hydrogen motion in chemical reactions.

## Methods

### Experimental Setup

In our experiments, a linearly polarized laser pulses with a center wavelength of 800 nm and a pulse duration of 40 fs were produced using a Ti:sapphire laser with a repetition rate of 1 kHz. The laser beam was focused onto the molecular beam using a mirror with a focal length of 75 mm and its polarization was set to be vertical to the time-of-flight (TOF) direction. The base pressure of the main chamber and the dump chamber are $6 \times 10^{-7}$ Pa and $8 \times 10^{-8}$ Pa, respectively. The $C_2H_6$ molecules were introduced into the chamber through a supersonic expansion, causing the pressure of the dump chamber increases to $9 \times 10^{-8}$ Pa while the main chamber remains the same. The molecular ionization and fragmentation induced by the laser pulses were studied by the cold target recoil ion momentum spectroscopy (COLTRIMS)[33]. The produced electrons and ions were detected in coincidence by two time- and position-sensitive detectors through a uniform magnetic field provided by Helmholtz coils and a uniform electric field formed by a set of electrode plates, respectively, and their three-dimensional momentum vector can be extracted[50] (more details about the

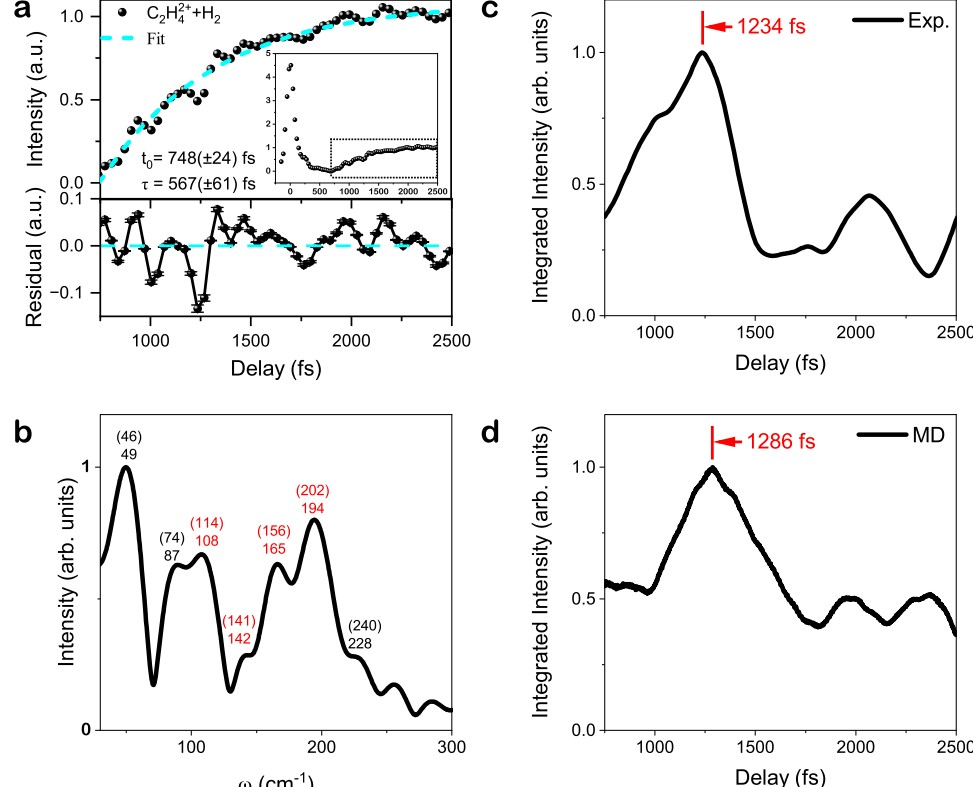

**Fig. 5 | Temporal dynamics of neutral H₂ formation dynamics. a** Time-resolved ion yields of $C_2H_4^{2+}$. An enhancement of ion yield was observed after a time delay of ~700 fs, as indicated by the dots in the inset. The main panel in a shows an enlarged and normalized view of the ion yield enhancement. The experimental data was fitted using an exponentially rising function. The rising time constant was determined to be 567($\pm$ 61) fs, and the initial instant time was found to be 748($\pm$ 24) fs. The oscillations in the ion yield were observed in both the raw data and the residual (lower sub-panel) obtained from the difference between the experimental data and the fitting curve. The errors of residual signals originate from the statistical uncertainties. **b** The Fourier transformed spectrum of the residual signal. The peaks labeled in red numbers were assigned to the vibrational mode frequencies (in brackets) of the molecular geometry around the CI. **c, d** The overlap between the time-resolved frequency distribution of the residual signal and velocity auto-correlation function of MD simulation with the characteristic frequency distribution of representative CI geometry (defined in Supplementary Note 2). The peak time for this overlap, corresponding to the characteristic time scale when $C_2H_6^+$ reaches the CI geometry, was determined to be 1234 fs and 1286 fs from the experiment and MD simulation, respectively.

experimental setup is described in Supplementary Note 1). The experimental laser intensity in Figs. 2 and 3 is 82 TW/cm², and we also carried the measurement at the intensity of 88 TW/cm² (see Supplementary) to validate the accuracy and reliability of the reconstruction method and to study the influence of laser intensity on the PMDs and branching ratios corresponding to the ground and excited cationic states. In the ion-electron coincidence measurement, for both laser intensities, the count rates for the ions were 0.075/pulse and 0.1/pulse, while the count rates for the electrons were 0.1/pulse and 0.15/pulse, respectively. The absolute experimental peak intensity is roughly estimated by comparing the double ionization yield of Xe with the reference curve first, with an error around 15%. Secondly, we further calibrated the intensity differences of the two conditions according to the measured pondermotive energy shift of ATI spectra of water molecules from the background. The accuracy of the relative intensity calibration in our two conditions is 6($\pm$1) TW/cm².

The time-resolved pump-probe measurements were used to track the non-adiabatic nuclear dynamics. A linearly polarized laser pulse with a central wavelength of 800 nm and a pulse width of 70 fs was split into two beams. One beam served as the pump pulse, which ionized ethane to its cationic form, and the other beam served as the probe pulse, which further ionized the parent ion or its fragments to the dicationic state. A translate stage was used to control the time delay between the pump and probe pulses. In this setup, the polarization of the pump pulse was kept parallel to the TOF axis while the polarization of the probe pulse was set perpendicular to the pump

pulse. The estimated intensities of the pump and probe pulses were 100 TW/cm² and 250 TW/cm², respectively. The three-dimensional momenta of the ions were collected using an ion imaging spectrometer in the COLTRIMS system. The ion count rates were measured to be 0.15/pulse and 0.25/pulse for the pump and probe pulses, respectively. The cross-correlation time of the pump-probe pulse is less than 100 fs.

## Data availability

The data that support the findings of this study are available within the Supplementary Material and upon request from the corresponding author. Source data are provided with this paper.

## Code availability

The codes used in this study are available from the corresponding author upon request.

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

## Acknowledgements

This study was supported by the National Natural Science Foundation of China (Grants Nos. 92261201 C.W., 12134005 D.D., 12274179 C.W., 12274273 X.H., 11874246 X.H., 12174009 Z.L., 12234002 Z.L., 92250303 Z.L.), and Beijing Natural Science Foundation (Grant No. Z220008 Z.L.).

## Author contributions

D.D. and C.W. conceived the experiments, Y.Y., X.L., Z.W., P.M., and K.D. conducted the measurements, Y.Y., M.Z., X.M., and M.L. prepared the figures. X.H., Z.L., H.R., M.Z., X.M., and S.Z. provided the calculations. W.L. and J.C. provided many valuable insights and suggestions. C.W., Y.Y., Z.L., X.H., and D.D. prepared the manuscript. All authors participated in the scientific discussions and in the revisions of the final manuscript.

## Competing interests

The authors declare no competing interests.
