## [Peer Review File · Nature Communications]

H₂ formation via non-Born-Oppenheimer hydrogen migration in photoionized ethaneREVIEWER COMMENTS

Reviewer #1 (Remarks to the Author):

The manuscript titled "H₂ formation photoionized ethane" by Yang et al. reports the proton migration in photo-ionized ethane to form an H₂ molecule. Using cold target recoil ion momentum spectroscopy, the authors have established the formation of H₂. The authors claim that the non-adiabatic coupling between ground and excited ionic states of ethane through conical intersection leads to a significantly high yield of neutral H₂ fragments. The authors also present theoretical calculations to support their results. The authors study the intensity dependence and study the pump-probe delay dependence to strengthen the findings.

Overall, the manuscript is well-written and presented well. I recommend publication.

Reviewer #2 (Remarks to the Author):

The manuscript NCOMMS-22-47713 "H₂ formation via non-Born-Oppenheimer hydrogen migration in photoionized ethane" by Yizhang Yang et al. represents a nice study of ultrafast molecular dynamics.

What are the noteworthy results?

This is a very nice piece of work. It beautifully applies the method of channel resolved ATI (Boguslavskiy et al. Science 335, 1336-1340 (2012)) to quantitatively resolve the pathways that contribute to the formation of free H₂ molecules from singly ionized Ethane. The authors find a surprisingly efficient production of H₂ which might have implications for interstellar chemistry. The authors also perform a pump-probe experiment to study the evolution of the H₂ formation. Again surprisingly the H₂ formation occurs on much larger time scales (100s of fs) than the typical scales of proton motion (few fs).

Will the work be of significance to the field and related fields? How does it compare to the established literature? If the work is not original, please provide relevant references.

The work is original to the best of my knowledge. It adds to previous work in this space in that it combines photoelectron and ion spectra for a more complete picture. The slow H₂ formation is a prime example for de-excitation through a conical intersection and conversion of electronic energy into rovibrational energy. Conical intersection dynamics has generated a lot of interest in recent years due to its relevance for light harvesting, and molecular resistance against radiation damage, to name a few.

Does the work support the conclusions and claims, or is additional evidence needed?

I do not doubt that the presented evidence supports the conclusions. However, I do have two questions:

First of all, I am wondering if the momentum distribution of the ion fragments can add anything. The authors are using a COLTRIMS which measures also ion momentum distributions, yet it seems they use the ion-detector mainly as a TOF mass spectrometer. It is surprising that the modeling of the molecular orbital ionization works so well, considering that the molecules are presumably randomly aligned. The authors should address whether Freeman resonances or at least the HOMO vs HOMO-1 have any alignment dependence. Furthermore, in their time-resolved measurement the authors could have really leveraged their technique, especially since the probe pulse was ionizing. Why only measure the C₂H₄(2+)+H₂ yield? Why not also the C₂H₄(+) + H₂(+) and look at the KER?

This brings me to my second point: I felt the discussion of the temporal evolution was somewhat unsatisfying. Maybe the authors can improve on the explanation of the retrieval of the time it takes the ethane ion to arrive at the conical intersection. Is this not something that comes from the fit to the rise of the C₂H₄(2+) yield? More fundamentally where is the zero line of the C₂H₄(2+) channel? The authors claim an enhancement for large delay times, but it could also be a recovery to at large delays. That is, if the probe pulse comes late enough it only finds the C₂H₄(+) to ionize. At times closer to zero it will ionize an electronically excited Ethane ion which will take a different decay path such as C₂H₄(+) and H₂(+) or H(+) and H. It would be very instructive to know how these channels vary with

pump-probe delay. For that the authors would need to make use of their ion detector beyond a simple TOF spectrometer with multihit detection of ions, selecting for momentum correlation etc.

Are there any flaws in the data analysis, interpretation and conclusions? Do these prohibit publication or require revision?

I do not see any flaws in the data analysis, interpretation or conclusions. I find the writing at times cumbersome to the point of not being understandable and it surely needs a good copy editor to improve on its accessibility.

Why have the potential energy surfaces in Fig. 1 no axis labels or units on them? One could think they are just cartoons.

In Figure 5d) the results of the molecular dynamics simulation are labeled as "Cal." which is the same as in Fig 2 where it was used for the CCSFA calculations. In Fig. 4 the label "MD" was used.

Is the methodology sound? Does the work meet the expected standards in your field?

The methodology relies on previously published recipes, such as the reconstruction of the branching pathway by fitting ATI spectra. As such I judge the methodology sound. I am surprised though that the authors did not specify an error for the intensity, or describe how they determine the intensity. This is particular interesting as the authors report differences for measurements that vary less than 10% in intensity. Typically it is very difficult to specify the intensity very accurately experimentally. If the authors used their ATI-fit to determine the intensity then they should state it.

Also in the supporting material "I. Experimental Setup" the authors show a COLTRIMS setup with a supersonic jet, yet they state that the "...the sample gas pressure inside the mass

spectrometer was kept at 6×10^{-7} bar, approximately one order of magnitude higher than the typical base pressure." I am sure that simply by quoting this sentence the authors know what is wrong. So will leave it at that.

In summary, I must apologize for the delay in my report, and assuming that the above minor comments are being addressed I can probably recommend publication.

Reviewer #3 (Remarks to the Author):

Review of Manuscript by Yizhang Yang et al. titled "H₂ formation via non-Born-Oppenheimer hydrogen migration in photoionized ethane"

The manuscript reports on an interesting H₂ formation process through the photoionization of ethane (C₂H₆) molecules. The authors used a COLTRIMS reaction microscope to extract the channel-resolved 3D photoelectron momentum distribution (PMD) and reconstructed the PMD from different states. The reconstructed PMDs were found to have good agreement with those obtained using a CCSFA calculation method. While other papers have applied the same method (channel-resolved photoelectron spectrum using COLTRIMS) in H₂ for revealing the breakdown of Born-Oppenheimer approximation (e.g. Phys. Rev. Lett. 118, 183201 (2017) by Mi et al.), this manuscript disentangled the two non-adiabatically coupled states that contribute to the H₂ formation in a more complex molecule (ethane), which is normally challenging in experiments. Moreover, using a pump-probe measurement, the formation time of neutral H₂ (~1200 fs) was extracted. This surprisingly long H₂ formation time implies a hydrogen migration process beyond the Born-Oppenheimer approximation.

The manuscript is well-structured, and all experimental and calculation results are convincingly and appropriately interpreted. Considering the significance of this process in other complex hydrocarbon molecules, I think this manuscript can be published in Nature Communications. However, the manuscript also has some errors and unclear parts.

Therefore, the authors should consider the following questions and suggestions to improve the manuscript:

1. In the “Methods-Experimental Setup” and Supplementary material, the authors wrote “the sample gas pressure inside the mass spectrometer was kept at 6×10^{-7} bar...” For the COLTRIMS spectrometer, this pressure is a few orders higher than normal. At this pressure, it is not possible to perform a valid coincidence experiment with COLTRIMS. Even if the pressure was 6×10^{-7} mbar, it is still too high, especially for channel-resolved photoelectron measurement. Where was the pressure measured, in the main chamber or in the dump?

2. What are the count rates of ions and electrons in the measurement? The author should mention them in the manuscript (probably in the methods section). Increasing count rate will lead to more false coincidences. This is particularly important for the $C_2H_4^+ + H_2 + e^-$ channel, as it is not possible to extract the electrons with momentum conservation conditions.

3. The authors should add numbers and units to the x and y axes of the potential energy surface plot (Fig. 1c).

4. Related to Point 3, could the authors briefly explain the g and h vectors of the potential energy surface in the main text? These two vectors are mentioned in the captions of Fig. 1 but not in the article. Also, there appears to be a typo in the last-second line of the caption. Should it be “g and h vectors” instead of “g and b vectors”?

5. The reconstruction of the PMD from the two states was not detailed in the main text. I suggest the authors add more details of the reconstruction processes, such as why it can be reconstructed based on the fitted branching ratio, how the two PMDs in Fig. 2(c) and (d) were reconstructed from those in Fig. 2(a) and (b)). Otherwise, readers might be confused even after checking the details in the supplementary materials.

6. The term “40/140 degree” in Fig. 3(a) is unclear. The author should either delete it from the figure or explain it in the caption.

7. Regarding the pump-probe experiment, the role of the probe pulses is not clearly described. The authors wrote “probe laser pulse monitors the H₂ formation dynamics by ionizing its partner ion C₂H₄⁺ to C₂H₄⁽²⁺⁾ at different time delays.” However, it is not clear why the increased yield of C₂H₄⁽²⁺⁾ could imply the H₂ formation time. Is the probe pulse a disruptive pulse that can prevent C₂H₄⁺ from being doubly ionized? Why?

To Referee A:

We are very grateful to Referee A for the positive evaluation of our work.

The manuscript titled "H₂ formation ... photoionized ethane" by Yang et al. reports the proton migration in photo-ionized ethane to form an H₂ molecule. Using cold target recoil ion momentum spectroscopy, the authors have established the formation of H₂. The authors claim that the non-adiabatic coupling between ground and excited ionic states of ethane through conical intersection leads to a significantly high yield of neutral H₂ fragments. The authors also present theoretical calculations to support their results. The authors study the intensity dependence and study the pump-probe delay dependence to strengthen the findings. Overall, the manuscript is well-written and presented well. I recommend publication.

To Referee B:

We thank Referee B for the helpful comments. We have made modifications following the Referee's valuable suggestions. By adding new experimental evidences about the time-dependent dynamics of neutral H₂ formation and other revisions as the referee suggested, we believe the present version of the manuscript can well address all the comments from the referee.

The manuscript NCOMMS-22-47713 "H₂ formation via non-Born-Oppenheimer hydrogen migration in photoionized ethane" by Yizhang Yang et al. represents a nice study of ultrafast molecular dynamics. What are the noteworthy results? This is a very nice piece of work. It beautifully applies the method of channel resolved ATI (Boguslavskiy et al. Science 335, 1336-1340 (2012)[1]) to quantitatively resolve the pathways that contribute to the formation of free H₂ molecules from singly ionized Ethane. The authors find a surprisingly efficient production of H₂ which might have implications for interstellar chemistry. The authors also perform a pump-probe experiment to study the evolution of the H₂ formation. Again surprisingly the H₂ formation occurs on much larger time scales (100s of fs) than the typical scales of proton motion (few fs). Will the work be of significance to the field and related fields? How does it compare to the established literature? If the work is not original, please provide relevant references. The work is original to the best of my knowledge. It adds to previous work in this space in that it combines photoelectron and ion spectra for a more

complete picture. The slow H_2 formation is a prime example for de-excitation through a conical intersection and conversion of electronic energy into rovibrational energy. Conical intersection dynamics has generated a lot of interest in recent years due to its relevance for light harvesting, and molecular resistance against radiation damage, to name a few. Does the work support the conclusions and claims, or is additional evidence needed? I do not doubt that the presented evidence supports the conclusions. However, I do have two questions:

Comment 1(a): *First of all, I am wondering if the momentum distribution of the ion fragments can add anything. The authors are using a COLTRIMS which measures also ion momentum distributions, yet it seems they use the ion-detector mainly as a TOF mass spectrometer. It is surprising that the modeling of the molecular orbital ionization works so well, considering that the molecules are presumably randomly aligned. The authors should address whether Freeman resonances or at least the HOMO vs HOMO-1 have any alignment dependence.*

Reply 1(a): As mentioned by the referee, the alignment of molecules plays a crucial role in strong field ionization. The simulated energy spectra and photoelectron momentum distributions (PMDs) presented in the manuscript are the averaged results obtained from different orientations, aiming to mimic the experimental conditions where molecules are assumed to be randomly aligned. Figs. R1 and R2 illustrate the simulated ionization yield and PMDs for the 2E_g (HOMO orbital) and ${}^2A_{1g}$ (HOMO-1 orbital) states of ethane vs the angle between the C-C axis and the laser polarization. According to our calculations, the ionization yield of the 2E_g state exhibits a strong dependence on the orientation angle, with minima observed at 0 and 90 degrees, and a maximum at 45 degrees. Consequently, the orientation-averaged results predominantly reflect the ionization of molecules oriented around 45 degrees. On the other hand, the ionization yield of the ${}^2A_{1g}$ state shows a weaker dependence on the orientation angle. This contrasting behavior arises from the different compositions of their wave functions. In the case of the 2E_g state, ionization is primarily influenced by the contribution of the C-C atomic pair, hence its sensitivity to orientation. However, for the ${}^2A_{1g}$ state, both the C-C pair and H-H pairs contribute to ionization. The presence of three H-H pairs oriented in different directions mitigates the orientation effect.

Furthermore, we observe that the PMD patterns for both the 2E_g and ${}^2A_{1g}$ states display

FIG. R1: The calculated ionization yield and photoelectron momentum distributions (PMDs) for the 2E_g state of ethane vs the angle θ between the C-C axis and the laser polarization are presented in (a) - (h).

FIG. R2: The same results as Fig. R1 but for the ${}^2A_{1g}$ state of ethane.

a weak dependence on molecular orientation. The distinctive features used to differentiate between the two states, namely the fanning out structure for the 2E_g state and the arm-like structure for the ${}^2A_{1g}$ state, are prominent across nearly all orientation angles. This is primarily because the low-energy portion of the PMD pattern is primarily governed by the symmetry of the molecular orbital, which remains preserved regardless of orientation. Based on our calculations, we conclude that the alignment of the molecules has a negligible impact on the reconstructed results of the two states under the specific conditions considered in this study.

Changes: We added Supplementary Figures 8 and 9 and the related discussions about

the alignment in the supplementary information. A sentence "where the contributions from different molecular orientations are averaged" is added in the caption of the Figure 2 of the main text.

Comment 1(b): *Furthermore, in their time-resolved measurement the authors could have really leveraged their technique, especially since the probe pulse was ionizing. Why only measure the $C_2H_4^{2+}+H_2$ yield? Why not also the $C_2H_4^++H_2^+$ and look at the KER?*

Reply 1(b): In our time-resolved measurement, the neutral H_2 formation channel $C_2H_6^+ \rightarrow C_2H_4^+ + H_2$ is triggered by the pump pulse, while the probe pulse is used to monitor its formation dynamics by ionizing the transient $C_2H_4^+$ to $C_2H_4^{2+}$ (i.e, $C_2H_4^+ + H_2 \rightarrow C_2H_4^{2+} + H_2$ (1)). As pointed out by the referee, the probe pulse can also ionize H_2 to H_2^+ , allowing for the monitoring of the H_2 formation dynamics by detecting the delay time-dependent dynamics of the ion pair ($C_2H_4^+ + H_2^+$). Based on the ion-ion coincidence technique with COLTRIMS, we obtained the delay time-dependent yield and kinetic energy release (KER) of the two-body Coulomb explosion channel $C_2H_4^+ + H_2 \rightarrow C_2H_4^+ + H_2^+$ (2). The yield vs KER and delay-time of this channel is presented in Fig. R3(a). A strong band with a KER of 3.8 eV is observed, showing no delay time dependence (see Fig. R3(b)). This band originates from the direct Coulomb explosion of $C_2H_6^{2+}$ [2], indicating that the dynamics of this pathway do not reflect the neutral H_2 formation dynamics. Interestingly, a weak band emerges after a time delay of approximately 700 fs, exhibiting strong delay-time dependence. The KER of this band decreases from 1.0 eV to 0.5 eV, and simultaneously, the yield of this band increases noticeably for time delays larger than 1000 fs (see Fig. R3(a) and (c)). These characteristics of the band effectively reflect the neutral H_2 formation dynamics. First, the appearance time of this band (approximately 700 fs) corresponds to the instant when the wave-packet reaches the conical intersection and the H_2 formation channel emerges, consistent with the results discussed in the main text for channel (1). Second, due to the large-scale motion of the hydrogen atom during the non-adiabatic dynamics along the cation, the KER of this band is smaller compared to the direct Coulomb explosion channel from the dication. As the delay time increases, the distance between the $C_2H_4^+$ and H_2 groups becomes larger, resulting in even smaller KER values (Fig. R3(c)). The neutral H_2 formation dynamics underlying the time-dependent features of the ($C_2H_4^+ + H_2^+$) channel is overall consistent with channel (1).

However, the yield of channel (2) is significantly lower than that of channel (1), so we mainly focused on channel (1) in the main text. The information about channel (2) is added in the supplementary information in the revised version.

FIG. R3: Delay time-dependent yield and kinetic energy release (KER) of the two-body Coulomb explosion channel $C_2H_6^{2+} \rightarrow C_2H_4^+ + H_2^+$ are depicted in (a). (b) and (c) present the distributions of KER and ion-pair yields at different time delays for two different bands.

Changes: We added the delay time-dependent yield and KER of channel (2) as Supplementary Figure 10 in the supplementary information. We also added the discussions about this channel in the main text.

Comment 2: *This brings me to my second point: I felt the discussion of the temporal evolution was somewhat unsatisfying. Maybe the authors can improve on the explanation of the retrieval of the time it takes the ethane ion to arrive at the conical intersection. Is this not something that comes from the fit to the rise of the $C_2H_4^{2+}$ yield? More fundamentally where is the zero line of the $C_2H_4^{2+}$ channel? The authors claim an enhancement for large delay times, but it could also be a recovery to at large delays. That is, if the probe pulse comes late enough it only finds the $C_2H_4^+$ to ionize. At times closer to zero it will ionize an electronically excited Ethane ion which will take a different decay path such as $C_2H_4^+$ and H_2^+ or H^+ and H . It would be very instructive to know how these channels vary with pump-probe delay. For that the authors would need to make use of their ion detector beyond a simple TOF spectrometer with multihit detection of ions, selecting for momentum correlation etc.*

Reply 2: We sincerely appreciate the referee’s insightful comment. In order to determine the most probable arrival time of the initial H₂ formation wave-packet at the conical intersection (CI), we conducted an analysis of the coherent vibrational motion and derived their characteristic frequencies. By comparing the time-dependent evolution of the measured frequencies with the frequencies associated with the representative CI geometry, we found that the majority of the wave-packet arrives at the CI within the time range of 1200-1300 fs.

Additionally, the temporal evolution of the neutral H₂ formation channel can be monitored through the delay-time dependent yields of the (C₂H₄²⁺ + H₂) channel. The rise of C₂H₄²⁺ as a function of time delay begins at approximately 700 fs, indicating the instant of the nuclear wavepacket starts to arrive to the conical intersection. This appearance time is consistent for both the (C₂H₄²⁺ + H₂) channel and the (C₂H₄⁺ + H₂⁺) channel. By fitting the increasing yields of C₂H₄²⁺, we determined a rising time constant of 567 (±61) fs, which represents the decay time constant of the entire initial wave-packet that exponentially decays via leaking through the CI. Therefore, we can conclude that the formation time of neutral H₂ is approximately 700 fs + 567 fs = 1267 fs after the initial ionization. Both methods yield consistent conclusions, suggesting that the majority of the initial wave-packet reaches the CI at approximately 1300 fs.

Moreover, at early times before the wave-packet reaches the CI, as the referee anticipated, we discovered an alternative channel: C₂H₆⁺ → CH₃⁺ + CH₃ (3), exhibiting opposite time-dependent dynamics compared to channel (1) (C₂H₄²⁺ + H₂), as shown in Fig. R4. The dynamics of (C₂H₄⁺ + H₂⁺) channel has been discussed in the reply to the Comment 1 and (C₂H₄⁺ + H₂⁺ + H) channel has not been observed in our measurement, thus we mainly focus on the Channel (3) here. Channel (3) originates from the dissociation of higher excited states of the cation, displaying a clear enhancement from 500 fs and subsequently decreasing after 1300 fs. Furthermore, the increasing counts of channel (3) and the decreasing counts of channel (1) are observed at similar levels, indicating that the excitation of C₂H₆⁺ serves as an dominant pathway before the wave-packet arrives at the CI. After passing through the CI, the ionization of C₂H₄⁺ becomes more significant relative to channel (3). This result suggests that the initial wave-packet, after ionization, can be promoted to higher excited states by the probe pulse before the reformation of C₂H₄⁺.

FIG. R4: The delay time-dependent yields of two different channels. Their yields exhibit opposite trend as increasing the time delay. Two curves are normalized to their minimum values.

Changes: The comparison of the delay time-dependent yields of channel (1) and channel (3) is added as Supplementary Figure 11, and the descriptions about the probe pulse-induced excitation to higher excited states are also included.

Comment 3: *Are there any flaws in the data analysis, interpretation and conclusions? Do these prohibit publication or require revision? I do not see any flaws in the data analysis, interpretation or conclusions. I find the writing at times cumbersome to the point of not being understandable and it surely needs a good copy editor to improve on its accessibility.*

Reply 3: We asked the professional copy editor to revise the writing of the manuscript to improve the readability.

Comment 4: *Why have the potential energy surfaces in Fig. 1 no axis labels or units on them? One could think they are just cartoons.*

Reply 4: We thank the referee for pointing out this issue. We would like to clarify that

Figure 1a in the manuscript is solely a schematic diagram intended to illustrate the bound and dissociative characteristics of the 2E_g and ${}^2A_{1g}$ states, respectively, for the H_2 formation reaction of $C_2H_6^+$. The potential energy surface around the representative conical intersection is shown in Figure 1c. To address this concern, in the revised manuscript, we have removed the “eV” label from the y -axis of Figure 1a to prevent any possible misunderstanding. Furthermore, we have added appropriate numbers and units to both the x and y axes of Figure 1c. The revised figure (see Fig. R5), is updated in the main text.

FIG. R5: A revised schematic diagram for the H_2 formation reaction of $C_2H_6^+$, contributed both via direct dissociation from excited ${}^2A_{1g}$ state and non-adiabatic coupling from ground 2E_g state via conical intersection (CI). (b) Schematic diagram for tunneling ionization and Freeman resonance. (c) The potential energy surface around the representative CI.

Changes: We revised Figure 1 in the main text accordingly.

Comment 5: In Figure 5d) the results of the molecular dynamics simulation are labeled as “Cal.” which is the same as in Fig 2 where it was used for the CCSFA calculations. In Fig. 4 the label “MD” was used.

Reply 5: We changed the label “Cal” in Figure 5d to MD for consistence. MD stands for the molecular dynamics simulation.

Comment 6: *Is the methodology sound? Does the work meet the expected standards in your field? The methodology relies on previously published recipes, such as the reconstruction of the branching pathway by fitting ATI spectra. As such I judge the methodology sound. I am surprised though that the authors did not specify an error for the intensity, or describe how they determine the intensity. This is particular interesting as the authors report differences for measurements that vary less than 10% in intensity. Typically it is very difficult to specify the intensity very accurately experimentally. If the authors used their ATI-fit to determine the intensity then they should state it.*

Reply 6: As the referee pointed out, accurately determining the absolute peak intensity of the femtosecond laser pulse in strong-field physics is indeed challenging. We initially estimated the intensity by comparing the double ionization yield of Xe with a reference curve [3–5]. However, it is important to note that this method introduces an error in the intensity calibration, which can be larger than 15% [6]. The relative intensity difference between the two conditions can be defined by the measured energy offsets of above-threshold ionization (ATI). It is well-known that the energy of the ATI peak is given by the equation: $E = nh\nu - \text{IP} - U_p$, where IP represents the ionization potential and U_p denotes the pondermotive energy [7–9]. U_p is directly proportional to the peak intensity. Thus, the difference between the two laser intensities in our case can be inferred from the measured pondermotive shifts of ATI between the two conditions. We extracted the ATI spectra of water molecule from the background, and obtained the pondermotive shift in the very same conditions as our measurements. The accuracy of the relative intensity calibration in our two conditions is $6(\pm 1)$ TW/cm². Due to the poor accuracy of the absolute intensity calibration, the relative yields shown in the right panel of Figure 4 only provide a qualitative estimation of the intensity-dependent ionization yield ratio of the $^2A_{1g}$ state relative to 2E_g states. Furthermore, we realized that this figure was not discussed in the main text as it is not directly relevant to the neutral H₂ formation dynamics. In the revised version of the manuscript, we have transferred this information to the supplementary information (Supplementary Table 5) to prevent any over-interpretation.

Changes: We provided the estimated error for the absolute intensity calibration in the caption of the Figure 4 in the main text, and the method we estimated the peak laser

intensity is included in the Methods section. The information included in the right panel of Figure 4 has been transferred to the supplementary information (Supplementary Table 5).

Comment 7: *Also in the supporting material "I. Experimental Setup" the authors show a COLTRIMS setup with a supersonic jet, yet they state that the "...the sample gas pressure inside the mass spectrometer was kept at 6×10^{-7} bar, approximately one order of magnitude higher than the typical base pressure." I am sure that simply by quoting this sentence the authors know what is wrong. So will leave it at that.*

Reply 7: We are sorry for this mistake, this part has been rewritten in the new version of the draft.

Changes: We revised this part in the Methods section as following: "The base pressure of the main chamber and the dump chamber are 6×10^{-7} Pa and 8×10^{-8} Pa, respectively.....the pressure of the dump chamber increases to 9×10^{-8} Pa while the main chamber remains the same. "

Comment 8: *In summary, I must apologize for the delay in my report, and assuming that the above minor comments are being addressed I can probably recommend publication.*

Reply 8: We thank the referee for the positive comments and valuable suggestions, which helped us to greatly improve the manuscript.

To Referee C:

We thank Referee C for the helpful comments. We have made modifications following to Referee's precious comment, which greatly help to improve the manuscript.

Comment 1: *The manuscript reports on an interesting H_2 formation process through the photoionization of ethane (C_2H_6) molecules. The authors used a COLTRIMS reaction microscope to extract the channel-resolved 3D photoelectron momentum distribution (PMD) and reconstructed the PMD from different states. The reconstructed PMDs were found to have*

good agreement with those obtained using a CCSFA calculation method. While other papers have applied the same method (channel-resolved photoelectron spectrum using COLTRIMS) in H_2 for revealing the breakdown of Born-Oppenheimer approximation (e.g. *Phys. Rev. Lett.* 118, 183201 (2017) by Mi et al.[10]), this manuscript disentangled the two non-adiabatically coupled states that contribute to the H_2 formation in a more complex molecule (ethane), which is normally challenging in experiments. Moreover, using a pump-probe measurement, the formation time of neutral H_2 (1200 fs) was extracted. This surprisingly long H_2 formation time implies a hydrogen migration process beyond the Born-Oppenheimer approximation.

The manuscript is well-structured, and all experimental and calculation results are convincingly and appropriately interpreted. Considering the significance of this process in other complex hydrocarbon molecules, I think this manuscript can be published in *Nature Communications*. However, the manuscript also has some errors and unclear parts. Therefore, the authors should consider the following questions and suggestions to improve the manuscript:

Reply 1: We thank the referee for providing a more perfect references that clearly explains the basis and source for our experimental approach. We cite them in the main text of the revised manuscript and summarize these important previous works.

Comment 2: In the “Methods-Experimental Setup” and Supplementary material, the authors wrote “the sample gas pressure inside the mass spectrometer was kept at 6×10^{-7} bar” For the COLTRIMS spectrometer, this pressure is a few orders higher than normal. At this pressure, it is not possible to perform a valid coincidence experiment with COLTRIMS. Even if the pressure was 6×10^{-7} mbar, it is still too high, especially for channel-resolved photoelectron measurement. Where was the pressure measured, in the main chamber or in the dump?

Reply 2: We thank the referee for pointing out this mistake, without molecular beam, the pressure for the main chamber is 6×10^{-7} Pa, and the pressure in the dump is 8×10^{-8} Pa. With molecular beam, the pressure for the main chamber stays constant, and the pressure in the dump is approximately 9×10^{-8} Pa.

Changes: We revised this part in the Methods section as following: “The base pressure

of the main chamber and the dump chamber are 6×10^{-7} Pa and 8×10^{-8} Pa, respectively.....the pressure of the dump chamber increases to 9×10^{-8} Pa while the main chamber remains the same. ”

Comment 3: *What are the count rates of ions and electrons in the measurement? The author should mention them in the manuscript (probably in the methods section). Increasing count rate will lead to more false coincidences. This is particularly important for the $C_2H_4^+ + H_2 + e^-$ channel, as it is not possible to extract the electrons with momentum conservation conditions.*

Reply 3: The counts rates for our measurement are listed in the following Table 1. In the ion-electron coincidence measurement, for the two laser intensities, the count rates for the ions are 0.075/pulse and 0.1/pulse, and for the electrons are 0.1/pulse and 0.15/pulse, respectively. For the pump-probe measurement, the ion-ion coincidence mode is used, the count rates are 0.15/pulse and 0.25/pulse for the pump and probe pulse, respectively.

Count rates (/pulse)	82 TW/cm ²	88 TW/cm ²	pump	probe
ion	0.075	0.1	0.15	0.25
electron	0.1	0.15	N/A	N/A

TABLE I: The count rates of ions and electrons in single pulse which originates from the laser system with 1 kHz frequency.

Changes: The count rates are summarized in the Methods section.

Comment 4: *The authors should add numbers and units to the x and y axes of the potential energy surface plot (Fig. 1c).*

Reply 4: We thank the referee’ suggestion. In the revised manuscript, we add numbers and units to the x and y axes.

Changes: The Fig. 1c has been revised.

Comment 5: *Related to Point 3, could the authors briefly explain the g and h vectors of the potential energy surface in the main text? These two vectors are mentioned in the captions of Fig. 1 but not in the article. Also, there appears to be a typo in the last-second line of the caption. Should it be “g and h vectors” instead of “g and b vectors”?*

Reply 5: We thank the referee for this careful reviewing. In the revised manuscript, we add explanations about **g** and **h** vectors in the main text, and revised the typo.

Comment 6: *The reconstruction of the PMD from the two states was not detailed in the main text. I suggest the authors add more details of the reconstruction processes, such as why it can be reconstructed based on the fitted branching ratio, how the two PMDs in Fig. 2(c) and (d) were reconstructed from those in Fig. 2(a) and (b)). Otherwise, readers might be confused even after checking the details in the supplementary materials.*

Reply 6: We sincerely appreciate the referee for this valuable suggestion, and added the details about the procedure to reconstruction of the PMD in the main text. We also explained this procedure in more details in the Supplementary information.

Changes: For the paragraph “In order to disentangle the contributions from the two coupled states to the PMDs of the H₂ formation and parent ion channels, we firstly fit the angular-integrated ATI spectrum..... ” in the main text, we added more explanations about this procedure in the Supplementary note 4.

Comment 7: *The term “40/140 degree” in Fig. 3(a) is unclear. The author should either delete it from the figure or explain it in the caption.*

Reply 7: We delete “40/140 degree” in the figure and explain its meaning in the caption.

Comment 8: *Regarding the pump-probe experiment, the role of the probe pulses is not*

clearly described. The authors wrote “probe laser pulse monitors the H_2 formation dynamics by ionizing its partner ion $C_2H_4^+$ to $C_2H_4^{2+}$ at different time delays.” However, it is not clear why the increased yield of $C_2H_4^{2+}$ could imply the H_2 formation time. Is the probe pulse a disruptive pulse that can prevent $C_2H_4^+$ from being doubly ionized? Why?

Reply 8: We thank the referee for the valuable suggestions. In our pump-probe approach, the pump pulse ionizes neutral C_2H_6 to $C_2H_6^+$ and initiates non-adiabatic hydrogen formation dynamics, leading to the channel $C_2H_6^+ \rightarrow C_2H_4^+ + H_2$. This channel is known as a dominant and synchronized process for the formation of $C_2H_4^+$ and H_2 , and the appearance of the channel ($C_2H_4^+ + H_2$) (1) at longer delay times serves as an indicator for the formation of neutral H_2 after reaching the conical intersection. This observation is further supported by our experimental and theoretical vibrational frequency analysis based on the continuous wavelet transform analysis.

Additionally, we also measured the channel $C_2H_4^+ + H_2 \rightarrow C_2H_4^+ + H_2^+$ (2), where H_2 is ionized to H_2^+ by the probe pulse. This channel exhibits similar delay-time-dependent dynamics as the $C_2H_4^+ + H_2$ channel, including a consistent appearance time and increasing yield at larger time delays (see Fig. R3). This new information further confirms that the delay time-dependent dynamics of $C_2H_4^{2+}$ can reflect the neutral H_2 formation dynamics.

Finally, at early time delays before the initial wave-packet on the ground state or first excited state of the cation approaches the conical intersection, the probe pulse can excite this initial wave-packet to higher excited states of the cation, leading to the dissociation channel: $C_2H_6^+ \rightarrow CH_3^+ + CH_3$ (Fig. R4). We observed an increase in the ion yield of this channel at early delay times before 500 fs, followed by a decrease for larger time delays (greater than 1300 fs). This channel represents an dominant pathway before the wave-packet reaches the conical intersection.

Changes: We added more details about how we probed the H_2 formation dynamics. We also added Supplementary Figures 10 and Figure 11 to show this nonadiabatic dynamics by monitoring the channel (2) and channel (3).

-
- [1] Boguslavskiy, A. E. *et al.* The Multielectron ionization dynamics underlying attosecond strong-field spectroscopies. *Science* **335**, 1336–1340 (2012).
- [2] Kanya, R. *et al.* Hydrogen scrambling in ethane induced by intense laser fields: Statistical analysis of coincidence events. *J. Chem. Phys.* **136**, 204309 (2012).
- [3] Chaloupka, J. L. *et al.* Observation of a transition in the dynamics of strong-field double ionization. *Phys. Rev. Lett.* **90**, 033002 (2003).
- [4] Wang, C. *et al.* Dissociative double ionization of formic acid in intense laser fields. *Chem. Phys. Lett.* **496**, 32–35 (2010).
- [5] Sun, X. *et al.* Mechanisms of strong-field double ionization of Xe. *Phys. Rev. Lett.* **113**, 103001 (2014).
- [6] Quan, W. *et al.* Laser intensity determination using nonadiabatic tunneling ionization of atoms in close-to-circularly polarized laser fields. *Opt. Express* **24**, 23248 (2016).
- [7] Freeman, R. R. *et al.* Above-threshold ionization with subpicosecond laser pulses. *Phys. Rev. Lett.* **59**, 1092–1095 (1987).
- [8] Schell, F. *et al.* Sequential and direct ionic excitation in the strong-field ionization of 1-butene molecules. *Phys. Chem. Chem. Phys.* **20**, 14708–14717 (2018).
- [9] Wang, C. *et al.* Accurate *in situ* measurement of ellipticity based on subcycle ionization dynamics. *Phys. Rev. Lett.* **122**, 013203 (2019).
- [10] Mi, Y. *et al.* Electron-nuclear coupling through autoionizing states after strong-field excitation of H₂ molecules. *Phys. Rev. Lett.* **118**, 183201 (2017).

REVIEWERS' COMMENTS

Reviewer #2 (Remarks to the Author):

The authors of manuscript NCOMMS-22-47713 "H₂ formation via non-Born-Oppenheimer hydrogen migration in photoionized ethane" have addressed all of my questions very comprehensively and to my satisfaction.

I stumbled over phrasing in the sentence "Extensive researches have been conducted..." in the introduction, but this can be fixed by a good copy-editor.

Hence, I now can only recommend publication of the manuscript.

In view of a couple of very recent, relevant publications that use the same methodology I suggest that the authors add the following two papers to their reference list, if space allows:

Mi, Y., Wang, E., Dube, Z. et al. D₃⁺ formation through photoionization of the molecular D₂-D₂ dimer. Nat. Chem. (2023). <https://doi.org/10.1038/s41557-023-01231-z>

Zhou, L., Ni, H., Jiang, Z. et al. Ultrafast formation dynamics of D₃⁺ from the light-driven bimolecular reaction of the D₂-D₂ dimer. Nat. Chem. (2023). <https://doi.org/10.1038/s41557-023-01230-0>

Reviewer #3 (Remarks to the Author):

The authors have addressed my previous concerns convincingly and have corrected the mistakes highlighted by Referees B and C. The provided numbers in the Methods Section (Experiment Setup) are encouraging, demonstrating a high-quality coincidence measurement and the utilization of channel-resolved photoelectrons to elucidate the H₂ formation process in ethane photoionization. Moreover, the revised manuscript now includes additional details on the reconstruction of the photoelectron momentum distributions, both in the main text and supplementary material, enhancing its comprehensibility for readers. Based on these improvements, I recommend its publication in Nature Communications.

To Reviewer 2:

We thank the reviewer for the comment and positive attitude to our manuscript.

Comment: The authors of manuscript NCOMMS-22-47713 "H₂ formation via non-Born-Oppenheimer hydrogen migration in photoionized ethane" have addressed all of my questions very comprehensively and to my satisfaction. I stumbled over phrasing in the sentence "Extensive researches have been conducted..." in the introduction, but this can be fixed by a good copy-editor. Hence, I now can only recommend publication of the manuscript. In view of a couple of very recent, relevant publications that use the same methodology I suggest that the authors add the following two papers to their reference list, if space allows: Mi, Y., Wang, E., Dube, Z. et al. D₃⁺ formation through photoionization of the molecular D₂-D₂ dimer. Nat. Chem. (2023). <https://doi.org/10.1038/s41557-023-01231-z>. Zhou, L., Ni, H., Jiang, Z. et al. Ultrafast formation dynamics of D₃⁺ from the light-driven bimolecular reaction of the D₂-D₂ dimer. Nat. Chem. (2023). <https://doi.org/10.1038/s41557-023-01230-0>.

Reply: We rephrased the sentence "Extensive researches have been conducted..." to "Extensive researches have been conducted to elucidate the underlying mechanisms of the formation of ionic products, such as H₂⁺ and H₃⁺, where the dynamics of neutral H₂ play a key role" to improve the readability. And we have cited those two references suggested by the reviewer.

To Reviewer 3:

We are thankful to reviewer for agreeing to accept our manuscript.

Comment: We greatly appreciate the Reviewers 1 and 3's positive affirmations and supports for our work. The authors have addressed my previous concerns convincingly and have corrected the mistakes highlighted by Referees B and C. The provided numbers in the Methods Section (Experiment Setup) are encouraging, demonstrating a high-quality coincidence measurement and the utilization of channel-resolved photoelectrons to elucidate the H₂ formation process in ethane photoionization. Moreover, the revised manuscript now includes additional details on the reconstruction of the photoelectron momentum distributions, both in the main text and supplementary material, enhancing its comprehensibility for readers. Based on these improvements, I recommend its publication in Nature Communications.